# The Phylogeographic Diversity of EBV and Admixed Ancestry in the Americas–Another Model of Disrupted Human-Pathogen Co-Evolution

**DOI:** 10.3390/cancers11020217

**Published:** 2019-02-14

**Authors:** Alejandro H. Corvalán, Jenny Ruedlinger, Tomas de Mayo, Iva Polakovicova, Patricio Gonzalez-Hormazabal, Francisco Aguayo

**Affiliations:** 1Department of Hematology and Oncology, Pontificia Universidad Catolica de Chile, Santiago 8330034, Chile; jennyruedlinger@gmail.com (J.R.); tomasdemayo@gmail.com (T.d.M.); iva.polakovicova@email.cz (I.P.); 2Advanced Center for Chronic Diseases (ACCDiS), Pontificia Universidad Catolica de Chile, Santiago 8330034, Chile; 3Faculty of Sciences, School of Medicine, Universidad Mayor, Santiago 7510041, Chile; 4Program of Human Genetics, Instituto Ciencias Biomedicas, Faculty of Medicine, Universidad de Chile, Santiago 8380453, Chile; patriciogonzalez@uchile.cl; 5Department of Basic and Clinical Oncology, Faculty of Medicine, Universidad de Chile, Santiago 8380453, Chile; faguayo@med.uchile.cl

**Keywords:** gastric cancer, Epstein-Barr Virus, human ancestry, viral phylogeography

## Abstract

Epstein-Barr virus (EBV) is an etiological agent for gastric cancer with significant worldwide variations. Molecular characterizations of EBV have shown phylogeographical variations among healthy populations and in EBV-associated diseases, particularly the cosegregated BamHI-I fragment and XhoI restriction site of exon 1 of the *LMP-1* gene. In the Americas, both cosegregated variants are present in EBV carriers, which aligns with the history of Asian and European human migration to this continent. Furthermore, novel recombinant variants have been found, reflecting the genetic makeup of this continent. However, in the case of EBV-associated gastric cancer (EBV-associated GC), the cosegregated European BamHI-“i” fragment and XhoI restriction site strain prevails. Thus, we propose that a disrupted coevolution between viral phylogeographical strains and mixed human ancestry in the Americas might explain the high prevalence of this particular gastric cancer subtype. This cosegregated region contains two relevant transcripts for EBV-associated GC, the *BARF*-1 and miR-BARTs. Thus, genome-wide association studies (GWAS) or targeted sequencing of both transcripts may be required to clarify their role as a potential source of this disrupted coevolution.

## 1. Introduction

Since the recognition of the Epstein-Barr virus (EBV) as an etiological agent for gastric cancer (GC) [1], an explosion of research has taken place, focused on the pathogenesis and novel therapeutic developments for this particular subtype (for a review in this series see Reference [2]). In this scenario, meta-analyses and aggregated individual-level studies have shown significant variations in the worldwide rate of EBV-associated GC with a particularly high prevalence in the Americas compared with those reported in the Asian, European, and African continents [3,4,5,6,7]. Furthermore, studies in the so-called New World indicate that European (mostly Spanish) and Amerindian admixture (i.e., Hispanic ancestry) is clearly associated with higher rates of EBV infection in GC when compared with rates from those with White/non-Hispanic heritage in the US or Brazilians with Japanese ancestry (Table 1) [8,9,10,11]. Thus, the Americas provide a unique opportunity for uncovering the molecular basis of this particular subtype of GC. Here, we propose that a disrupted EBV-human co-evolution based on the combination of phylogeographic polymorphisms of EBV and mixed ancestries in the Americas might explain the high prevalence of this particular disease subtype.

## 2. Phylogenetic Classification of EBV

The molecular characterization of EBV has been facilitated by whole genome sequences, restriction fragment length polymorphisms (RFLP), genome-wide association studies (GWAS) from healthy donors, the study of benign and malignant lesions, and naturally infected GC SNU-719 cell lines [12,13,14,15,16] (for a review see Reference [17]). EBV can be classified as type 1 or 2, based on the substitution of 1.8 kb in the C-terminal domain of the *EBNA*-2 gene. Nucleotide differences at *EBNA*-3A, -3B, and -3C genes also contribute to this classification [18]. These two types of EBVs display phylogeographical differences as the EBV type-1 is the most common strain in the Asian, European, and American continents, whereas type-2 is frequently found in Africa [19,20]. These subtypes differ in their capacity to transform B-lymphocytes into a proliferative state [21]. A second classification of EBV has been elaborated by the RFLP map of the prototype EBV B95-8 genome after digestion with the BamHI restriction enzyme (Figure 1). The BamHI-F fragment is found in the majority of the healthy population and in EBV-associated diseases in Europe, Africa, and the Americas, including EBV-associated GC [22,23]. The presence of an extra BamHI restriction site within this fragment is the ‘‘f’’ variant. Although initially identified in nasopharyngeal carcinomas (NPC) (an undifferentiated epithelial-like tumor originating in the pharynx) [24,25], subsequent reports have found it in a low frequency among the healthy population and in EBV-associated diseases worldwide [23,26]. The “f” variant is located in the promoter region of the *EBNA*-1 gene, however, the functional significance of this polymorphism is currently unknown.

The BamHI–I fragment harbors one of the greatest ranges of phylogeographical variations among healthy donors and EBV-associated diseases [23,26,27]. This fragment predominates in Asia, while the presence of an extra BamHI site defines the type “i”, which prevails in Europe and Africa [25,28,29,30,31]. Another relevant phylogeographical variation in the EBV chromosome is the polymorphism of the XhoI restriction site of exon 1 of *LMP-1* gene. In Asia, the predominant viral strain lacks this restriction site (i.e., XhoI loss) [32]. However, the presence of this site (i.e., XhoI) defines the European and African subtypes [33].

Since these two sites (the BamHI–I fragment and the XhoI restriction sites) are closely located in the viral genome (see Figure 1), cosegregation is found in the Asian as well as the European and African strains. In the Americas, these cosegregations are also present among EBV carriers at a population level aligned with the waves of human migration [23,34]. In addition, novel recombinant variants have been found in this continent [23,35]. The discovery of these strains in the Americas should be understood as an amalgam of fragments from ancestral EBV sequences reflecting the genetic makeup of this continent’s population [36,37,38].

## 3. The Molecular Structure of the Cosegregated BamHI–I Fragment and XhoI Region of the EBV

GWAS analysis found that latent genes were the most diverse regions of the viral genome with the *EBNA*-3A, -3B, -3C, *BPLF*-1 and *LMP-1* genes harboring the most abundant non-synonymous variants [41]. Interestingly, the *LMP-1* gene region which contains the XhoI polymorphisms involved in the phylogeographical classification of EBV, may contribute to the variations in the prevalence of EBV-associated GC throughout the world. Additionally, two relevant transcript regions are located between both cosegregated variants (the BamHI–I fragment and the XhoI restriction site) (Figure 1), the BamHI-A rightward transcripts (BARTs) and BamHI-A rightward frame-1 (*BARF*-1). BARTs transcripts have several distinct spliced forms [42,43], whereas *BARF*-1 is located downstream of BART and encodes 221 amino acids [43,44] which are translated into a protein of 31–33 kDa [45].

EBV-encoded *BARF*-1 is a putative viral oncogene (oncogenic initiator or oncogenic cofactor) in EBV-associated GC [46,47]. *BARF*-1 was shown to be expressed in tissues of various EBV-associated epithelioid malignancies, but not in those of lymphoid malignancies. Using a specialized *BARF*-1-nucleic acid sequence-based amplification assay (NASBA), it has been demonstrated that *BARF*-1 exists in all EBV-associated GC tissues [46,48]. Furthermore, it was shown that recombinant expression of *BARF*-1 induced tumorigenic transformation of mouse fibroblasts and tumor formation in newborn rats [49]. Reconstitution of an NPC-type EBV infection using NPC-derived cell lines demonstrated that *BARF*-1 contributes to the tumorigenicity of NPC cells [50]. *BARF*-1 enhances the tumorigenicity of EBV-negative B-lymphocyte-derived cell lines [51,52], inhibits apoptosis by activating *bcl*-2 [53], and induces cell cycle activation [54,55]. *BARF*-1 has sequence homology with the human colony stimulating factor-1 receptor (CSF-1), which is the gene product of the human proto-oncogene, c-*fms* [56]. It has been suggested that CSF-1 and its receptor are involved in the tumorigenicity of epithelial cells, as increased expression is observed in GC as well as other carcinomas [50,57,58]. *BARF*-1 is secreted by EBV-carrying B cells upon the induction of lytic infection and it binds to CSF-1, inhibiting the binding of CSF-1 to the CSF-1 receptor. This leads to inhibition of IFN-α secretion and modulates the fates of immune-related cells such as macrophages [56,59,60]. Secreted *BARF*-1 can upregulate nuclear factor κB (NFκB) in an autocrine and paracrine manner in GC [47]. *BARF*-1-expressing GC cells displayed a high rate of proliferation, high levels of NFκB, and miR-146a, which can be reversed by *NF*κB knockdown [47]. Silencing *BARF*-1 upregulates the expression of pro-apoptotic proteins and downregulates the expression of anti-apoptotic proteins [61].

In the BART region, a cluster of 22 miRNAs precursors have been described [62]. These miRNAs generate 40 out of 44 mature miRNAs encoded by EBV (Figure 1) [62,63,64]. miR-BARTs show higher overall expression levels in EBV-infected epithelial cancers in comparison to EBV-infected lymphoblastoid cell lines and Burkitt’s lymphoma [65]. It is well established that these viral miRNAs modulate the host inflammatory response and favor EBV evasion, facilitating the maintenance of the latent infection and contributing to carcinogenesis [66,67]. These findings propose that miR-BARTs are key players in epithelial malignancies such as EBV-associated GC [65]. A comprehensive review of miR-BARTs and their function in EBV-associated GC has recently been published [68]. Taken together, it seems plausible that not only coding genes such as *BARF*-1, but also noncoding genes (i.e., miR-BARTs), may act as a potential source of variability in EBV-associated GC. Therefore, GWAS studies of the EBV genome, as well as targeted sequencing of the *BARF*-1 and BARTs transcripts in EBV-associated GC and the healthy population, will be essential for expanding our understanding of viral diversity.

## 4. Human Ancestry in the Americas and EBV-Associated Gastric Carcinoma

The complex demographic diversity of the Americas stems from many different waves of migration. In this regard, three major ethnicities contributed to the genetic makeup of this population: Amerindians, Europeans, and Africans [34,69,70]. A variety of approaches have been used to estimate the complexity of genetic ancestry in the Americas tracing back 15,000 years to the first waves of Asian-derived Amerindian migrations across Beringia [34,71,72]. This was followed by European migration during the mid-sixteenth century, initiating the genetic mixing of these populations [73]. The African component was primarily introduced through the slave trade during the seventeenth century, adding more complexity to the demographic diversity of the Americas [34].

Thus, the so-called New World provides a unique opportunity for uncovering the genetic basis of diseases [37,70]. In particular, differences in the incidences and mortality rates of GC according to ancestry have shown a disproportionate burden within the Amerindian populations [74,75]. Specific examples are the indigenous Inuit and Mapuche populations [76] who inhabit the Northern Arctic regions and Patagonia, respectively. In the case of the Inuit, the incidence of GC has increased significantly in recent decades despite a decline in the global mortality rate during the same time period [77,78,79,80,81]. Mapuche ancestry appears to be a major risk factor for GC in Chile, a country with one of the highest age-standardized incidence rates of the disease worldwide (23/100,000 inhabitants) [82]. Other examples include Peru and Colombia, where a positive association between Amerindian ancestry and GC has also been described [34,83,84]. In Brazil, da Silva and colleagues [85] reported the occurrence of GC in the Amazon region, which features an admixed population. Using a case-control design and multiple logistic regression analysis, these authors found that for every 10% increase in European ancestry, there is a 20% decrease in the probability of developing GC (*p* = 0.01; OR = 0.81; 95% CI 0.54–0.88), suggesting that European ancestry may be a protective factor for this disease. This information correlates with the fact that European countries tend to report lower incidence rates of GC than Asia or the Americas [82]. Thus, these examples highlight the major role of Amerindian ancestry in the occurrence of GC.

In the case of EBV-associated GC, the combination of phylogeographic polymorphisms of EBV and mixed ancestries in the Americas requires further exploration. As shown in Figure 2, in a healthy population, the human admixed heritage in Chile reflects European (48.5%) and Asian-derived Amerindian (49.9%) with a minor African component (1.6%) [37].

Accordingly, the EBV recombinant strains prevail (41.3% and 28.6%) and the cosegregated ancestral strains are found at a lower frequency (17.5% Asian and 12.7% European) [23]. However, in Peru, a country with a predominantly Amerindian population (75% Amerindian vs. ~25% European) [34], the Asian strain predominates (79.0%) with a small proportion of European and recombinants variants (8.6%, 7.6% and 4.8%, respectively) [35]. In the case of EBV-associated GC, the cosegregated European BamHI-”i” fragment and XhoI restriction site strain prevails in Chile and Peru (100% and 54.5%, respectively) [23,35]. The predominance of the European strain could be likely due to particular recombinations of genes located within the BamHI fragment and XhoI region–as is the case of *BARF*-1 and BARTs in an Amerindian ancestry host. The lack of ancestral coadaptation supports the proposal of a disrupted co-evolution in the case of EBV-associated GC.

## 5. Other Examples of “Disrupted Co-Evolution” in Cancer-Related Infectious Agents

The interaction between the phylogeography of EBV and human ancestry is not a unique feature of this virus. A disrupted co-evolution has been previously proposed for *H. pylori* and human papillomavirus (HPV) and their associated tumors (GC and cervical cancer, respectively) (for a review see [86]). *H. pylori*, a bacterium that chronically colonizes the gastric mucosa, coevolved with human migration patterns [87]. In the Americas, genomic data has revealed the rapid evolution of *H. pylori* over the last 500 years. Phylogenetic studies based on genomic strain data have shown that most of the strains cluster according to their country of origin, suggesting that subpopulations of *H. pylori* have evolved at an accelerated rate, in order to adapt to particular human ancestries [88,89]. This rapid adaptation has been associated with three genes encoding outer membrane proteins which are important for the attachment of the bacterium to the gastric mucosa [88]. This scenario might account for the regional variability of GC [88,89]. This observation was originally assessed by de Sablet et al. [90]. These authors analyzed the phylogeographic origin of *H. pylori* isolates from two locations in Colombia with strikingly different incidences of GC: Tuquerres, in the Andes mountains (150/100,000 inhabitants) and Tumaco, on the coast (6/100,000 inhabitants). Using Multi-Locus Sequence Typing (MLST), these authors found that 100% of the isolates from Tuquerres but only 34% from Tumaco were classified as hpEurope. In the latter location, the remaining 66% isolates were hpAfrica1. This finding mirrored the ethnic composition of the host in both locations, being admixture European-Amerindian ancestry in Tuquerres and African-European ancestry in Tumaco. Functional studies have shown that differential expression between the European and African strains is observed in virulence factors, such as *cagA*, *vacA*, and *babB* and were associated with increased gastric histologic lesions in human gastric samples [90].

HPV is a naked circular double-stranded DNA virus with more than 200 genotypes based on the genome sequence of *L1* gene [91]. High-risk (HR) strains, HPV-16 and -18 are the most frequent HPV detected in cervical, anogenital, and some head and neck cancers. Subtypes and variants of HPV-16 cluster into five major branches of a phylogenetic tree: European (E), Asian/American (AA), East Asian (As), and two African (Af1 and Af2) [92,93]. In the case of HPV-18, subtypes and variants cluster into three major branches: African (Af), European (E), as well as Asian and American Indian (As + AI) [93]. A study of HR-HPV in an Italian cohort demonstrated that non-European variants of HPV-16, Af1 and AA, were found at an increased frequency in invasive lesions [94]. A separate study of female university students in the US showed that those infected with non-European HPV-16 variants were 6.5 times more likely to develop high-grade cervical intraepithelial neoplasia than those with European variants [95]. Based on the aforementioned observations found in *H. pylori* and HPV, Kodaman et al., [86] proposed the concept of “disrupted co-evolution” between the pathogen and its host as a contributor to the phylogeographic origin of disease. Here, we propose that this may also be the case for EBV-associated GC.

## 6. Are Phylogeographic Variations of Epstein–Barr Virus Relevant to Other EBV-Associated Diseases?

Disparities in the incidence of EBV-associated diseases, besides GC, vary greatly in different parts of the world [96]. EBV-associated epithelial cancers represent 80% of all EBV-associated malignancies; among these is NPC with an incidence of >120,000 new cases and >70,000 deaths [82]. NPC has a characterized geographical distribution, with a higher incidence in Southern China, Southeast Asia, and to a lesser extent, the Maghrebi Arabic regions of North Africa and the Northern Arctic [97]. Differences in the prevalence of different EBV type-1 and -2 strains, as well as BamHI-F region, have been observed in NPC [17]. Studies in Portugal, Hong Kong, and China reported that EBV type-1 and prototype F were the most prevalent [98,99,100]. Of note, the study from Portugal showed that type-2 and variant “f” were significantly associated with NPC (*p* = 0.019; RR = 8.90). The XhoI loss variant was present in most of the NPC cases [17]. It proved particularly high in countries such as China, Malaysia, and Taiwan with more than 80% of the cases presenting this variant [17,101,102,103]. On the contrary, the presence of the XhoI site is found in NPCs from North Africa [104], confirming the phylogeographic distribution of this polymorphism. Another variant of the *LMP-1* gene is the 30 bp-deletion (del-*LMP-1*) which seems to be more prevalent in NPC patients than healthy individuals in China, Malaysia, Hong-Kong, Taiwan, Tunisia, and Morocco [17,99,102,104,105,106,107]. However, in Portugal, 100% of NPC patients exhibited the wt-LPM1 variant [17]. A recent meta-analysis by da Costa et al. [105] confirmed the association between the 30-bp del-*LMP-1* and XhoI loss with NPC susceptibility, although they found no association when analyzing the cosegregation of these variants in NPC patients.

Hodgkin lymphoma (HL) is a disease that can also exhibit the presence of EBV. It presents two distinct disease entities, the commonly diagnosed classical Hodgkin lymphoma (CHL) and the uncommon nodular lymphocyte-predominant Hodgkin lymphoma [108,109]. EBV is found in only a proportion of CHL cases, but in tropical regions, up to 100% of the population is EBV-positive [108]. HL accounts for ~80,000 new cases and >26,000 deaths, according to a recent global report [82]. The prevalence of EBV in CHL differs according to age, sex, region, histologic subtype, and clinical stage, as confirmed by a meta-analysis [110]. According to this study, the reported prevalence of EBV infection in CHL was 47.9%, with a significantly higher rate in Africa, Central America, and South America. EBV-positive CHL showed a higher incidence in children than in adults and was also significantly related to male gender (OR = 1.8, 95% CI: 1.510–2.038; *p* < 0.001) [110]. Regarding the common EBV variants distribution for HL among populations, it appears that EBV type-1 is the most prevalent, which is similar to what was described for NPC [17]. Studies in China, Korea, Spain, Denmark, and Australia showed that EBV type-1 was the most prevalent type among HL patients [17,111,112,113,114,115]. Moreover, type-1 was also the predominant strain in EBV variants in South American CHL, accounting for 78% Argentine and 86% Brazilian cases [116]. In the case of the XhoI loss, this variant was found to be predominant in EBV-associated Hodgkin’s disease cases and in the healthy Chinese population [111] confirming the phylogeographic distribution of this polymorphism. For del-*LMP-1* variant distribution among populations, these are similar between China and Korea with more than 80% of the cases harboring the variant [17,111,113]. Similarly, in the case of South America, a higher frequency of the del-*LMP-1* variant was observed in lymphomas (65%) than in non-neoplastic controls (27%) (OR 4.97, CI 95% 1.53–16.79; *p* = 0.005) [116]. Although phylogeographic variants of EBV seems to be relevant also in other EBV-associated diseases, such as NPC and HL, the del-*LMP-1* variant has shown no differences in the case of EBV-associated GC [117].

## 7. Conclusions

Ancestral and recombinant strains of EBV in the Americas mirror the human genetic ancestry among the healthy population. However, there is a predominance of the European origin strain, based on the cosegregation of BamHI- I fragment and XhoI restriction site strain, in the case of EBV-associated GC. This observation proposes that a “disrupted co-evolution” might explain the high prevalence of EBV-associated GC in the Americas. Variations of two relevant transcripts, the *BARF*-1 and the miR-BARTs in this region might be associated with this high prevalence. Further studies are essential to expand our understanding of the phylogeographical diversity of EBV.

## Figures and Tables

**Figure 1 cancers-11-00217-f001:**
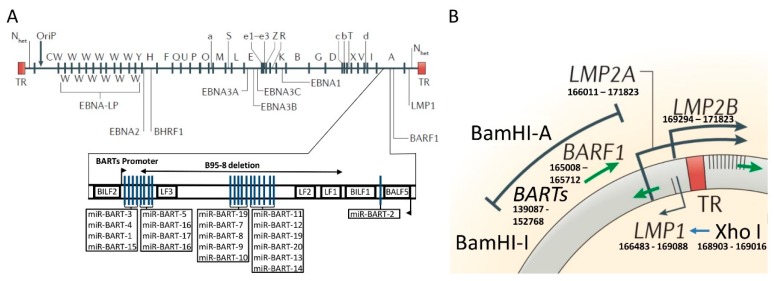
Genomic map of Epstein-Barr virus (EBV) genome with major phylogeographic variants. (**A**) Diagram shows the location of open reading frames for EBV latent proteins on the BamHI restriction map of the prototype EBV B95-8 genome. The BamHI fragments are named according to size and are indicated by capital letters, with A being the largest. Lowercase letters indicate variations in size within these fragments. TR refers to the terminal repeats at each end of the genome. Below is a schematic representation of viral miR-BARTs in sequence order with deletion indicated within the BamHI-A fragment of the EBV genome in the B95-8 strain. N_het_ is used to indicate heterogeneity in this region according to the number of TRs within different virus isolates. (**B**) Schematic representation of the genomic location of BamHI-I and BamHI-A fragments with TR. Within the exon 1 of *LMP1* gene is the XhoI restriction site. The region containing the BamHI-I and XhoI restriction site is 21,277 nucleotides long and contains several relevant transcripts (miR-BARTS and *BARF*-1) for EBV transformation abilities. Figure is adapted from References [39] (with permission to use part of the figure) and [40] (localization of miR-BARTs).

**Figure 2 cancers-11-00217-f002:**
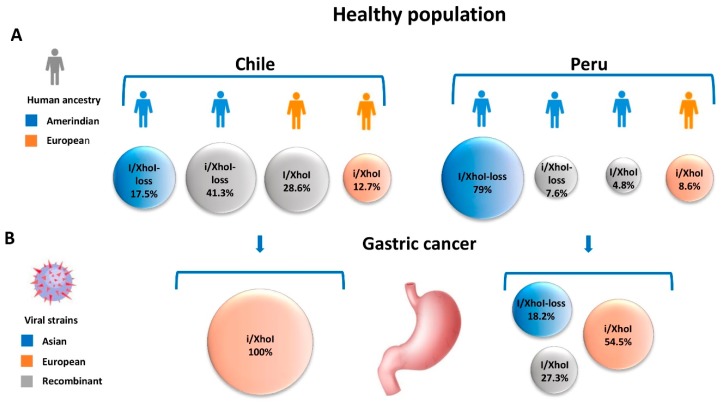
Human heritage and EBV strains in the healthy population and EBV-associated GC in Chile and Peru. (**A**) Human ancestries and Asian, European and recombinant viral strains identified in the healthy population of Chile and Peru. (**B**) Distribution and geographical origin of viral strains among EBV-associated GC patients in Chile and Peru, showing the predominance of the European BamHI-”i” fragment and XhoI restriction site strain. In the case of the healthy population, EBV strains were examined from throat washing specimens, and in the case of gastric cancer patients, from paraffin-embedded tumor specimens.

**Table 1 cancers-11-00217-t001:** Ancestry and Epstein-Barr virus-associated gastric cancer in the Americas.

Country	Ancestry	Sample Size	Rate of EBV Infection	*p* Value	References
Brazil	Hispanics	151	11.2%	0.01	[8]
	Japanese descendants	149	4.7%		
US	Mexican descendants	113	15.9%	0.023	[9,10]
	White/non-Hispanic	92	4.3%		

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
