# Peer review of "The Phylogeographic Diversity of EBV and Admixed Ancestry in the Americas–Another Model of Disrupted Human-Pathogen Co-Evolution"

_cancers, 2019, doi:10.3390/cancers11020217_

Round 1

Reviewer 1 Report

Cancers: Corvalan et al

This review article describes progress to date in associating EBV-related cancers to potentially oncogenic variants in selected regions of the EBV genome.  The authors propose how the combination of ancestral human genomics and EBV strains (based on polymorphisms in selected positions) might conspire to enhance risk of cancer progression across several types of EBV-related malignancy.  Analogous genomic and epidemiologic data from Hp and HPV models are also reviewed.  The topic is relatively new and this review serves as an important synthesis of work to date in understanding co-evolution of human and pathogen genomes in relation to tumorigenesis.

Suggestions to improve:

1.    Figure 1a.  The BAMH1 restriction map in figure 1a is a bit confusing.  The BAMH1 restriction map, first described a few decades ago, seems to name each EBV fragment from biggest to smallest, so A to Z for the biggest 26 fragments, and then continuing with lower case letters for the smaller and smaller fragments (e.g. a is 27th, b is 28th).  Clarify in figure 1 the polymorphism sites that you studied and that you confusingly also designate with lower case letters.   Be unambiguous about the sequence of the polymorphisms that were examined, by designating the position and the sequence change in the reference sequence (e.g. NC_007605 or an alternate reference sequence).  Instead of naming each polymorphism by a vague moniker such as “del-LMP1 variant”, name it using standard gene nomenclature such as “EBV NC_007605 (insert nucleotide position number within the reference sequence)(insert nucleotide change at that position such as G>A or del12)”.   Also predict if each DNA-based change results in amino acid change of the encoded protein(s), or in altered transcript sequence.  Modify figure 1 to display the correct nomenclature for each variant.  

2.    Line 87, remove the Nhet sentence since it is not in the figure below.

3.    Line 90, which two variants??  There is no indication of the position of the BAMH1-I polymorphisms in this figure.

4.    Line 99, do you mean between rather than within?

5.    Line 132, cite a reference if indeed this is “most relevant phylogeographic region of the EBV”.  Relevant for what?  Or are these sites simply the most examined portion of the EBV genome?  Describe GWAS studies that have comprehensively sequenced the entire EBV genome in disease and control groups, or if none then omit claims of relevance. 

6.    Figure 2.  It is not clear exactly what 4 genotypes are seen in the population from Chile.  Define each genotype unambiguously. 

7.    Figure 2, in GC patients, is the EBV strain shown the one found in tumors of the appx 10% of pts who have EBV positive cancer, or is it the strain found in their blood or saliva that might differ from the cancer strain? 

8.    The introduction describes a variant in the F (f) fragment but then does not further describe its association with cancer, therefore consider removing it.

9.    Consider replacing the term “co-segregation” by the genetics term “linkage disequilibrium”?

10. Line 199, do you mean “expression” as in RNA transcripts or do you mean genotype as in DNA variants?

Author Response

Rev 1

Suggestions to improve:

1. Figure 1a.  The BAMH1 restriction map in figure 1a is a bit confusing.  The BAMH1 restriction map, first described a few decades ago, seems to name each EBV fragment from biggest to smallest, so A to Z for the biggest 26 fragments, and then continuing with lower case letters for the smaller and smaller fragments (e.g. a is 27th, b is 28th).  Clarify in figure 1 the polymorphism sites that you studied and that you confusingly also designate with lower case letters.   Be unambiguous about the sequence of the polymorphisms that were examined, by designating the position and the sequence change in the reference sequence (e.g. NC_007605 or an alternate reference sequence).  Instead of naming each polymorphism by a vague moniker such as “del-LMP1 variant”, name it using standard gene nomenclature such as “EBV NC_007605 (insert nucleotide position number within the reference sequence)(insert nucleotide change at that position such as G>A or del12)”.   Also predict if each DNAbased change results in amino acid change of the encoded protein(s), or in altered transcript sequence.  Modify figure 1 to display the correct nomenclature for each variant. 

Answer:  We would like to thank the reviewer’s comments regarding this issue. Of note, it is important to address that Fig. 1 has been modified to fulfill the reviewer´s comments as best as possible. The lower-case letters does not refer to an specific variant, but rather to a region within the EBV genome that carries a loss or an extra BamHI site (Lung et al., Virology 1990). Therefore, the resulting “new” fragment is designated with a small letter (i.e “f” or “I”). This terminology is in accordance to the literature.  We acknowledge that the reviewer mentioned the reference sequence of EBV (NC_007605), where the whole genome of the EBV virus appears. Nevertheless, it is not possible to depict the exact position of an specific sequence variation, because in other to do so we must have the sequence files from all the studies cited (i.e FASTA files) and then align them to the reference genome or perform de novo assembly (Pasler et al., 2015 Journal of Virology). Moreover, when reviewing the EBV genome in the NBCI genome browser, variants are not described. This means that there is not a SNP database regarding the EBV genome. The terminology used in our manuscript is in accordance to the current literature. Since it is not possible to know exactly the genetic variants of the cited literature, we cannot predict the putative change in the protein sequence. We feel, with respect, that this requirement goes beyond the scope of this paper.

2. Line 87, remove the Nhet sentence since it is not in the figure below.

Nhet is part of the figure 1A, that is why it is explained in the figure legend.

3. Line 90, which two variants??  There is no indication of the position of the BAMH1-I polymorphisms in this figure.

This sentence of the legend Fig. 1 was rephrased (see lines 89-93).

4. Line 99, do you mean between rather than within?

Answer: We would like to thank the reviewer’s comments. We have changed to “between”

5. Line 132, cite a reference if indeed this is “most relevant phylogeographic region of the EBV”.  Relevant for what?  Or are these sites simply the most examined portion of the EBV genome? Describe GWAS studies that have comprehensively sequenced the entire EBV genome in disease and control groups, or if none then omit claims of relevance. 

Answer: We appreciated reviewer’s comment. We have deleted the phrase “both within the most relevant phylogeographic region of the EBV”. On the other hand, we appreciated reviewer´s comment about GWAS studies. However we would like to state that GWAS studies do not focus on full genome sequence (Bush and Moore, PloS Comput Biol 2012;8:e1002822). This type of studies is designed to examine the relationship between a genetic variant (SNP) and a phenotype using case-control design. Specific chips with thousands of probes detecting genetic variants are used (Low S-K et al, Clin Cancer Res 2014). Therefore, it is not possible to address the reviewer’s comment.

6. Figure 2.  It is not clear exactly what 4 genotypes are seen in the population from Chile.  Define each genotype unambiguously. 

Answer: We rephrase legend of Figure 2 clarifying Asian, European and recombinant viral strains identified in healthy population and EBV-associated GC from Chile and Peru (lines 169-171).

7. Figure 2, in GC patients, is the EBV strain shown the one found in tumors of the appx 10% of pts who have EBV positive cancer, or is it the strain found in their blood or saliva that might differ from the cancer strain? 

Answer: In the case of healthy population, EBV strains were examined from throat washing specimens, and in the case of gastric cancer patients, from paraffin-embedded tumor specimens. According to our paper (ref 84), the same strain was found, in both stomach and throat washings, in the case of EBV-associated GC.

8. The introduction describes a variant in the F (f) fragment but then does not further describe its association with cancer, therefore consider removing it.

Answer: We appreciate this reviewers’ comment. We have added further description on F(f) fragment in the section 6 lines 225-227. “EBV type-1 and -2 strains as well as BamHI-F region have been observed in NPC [17]. Studies in Portugal, Hong Kong, and China reported that EBV type-1 and prototype F were the most prevalent [96-98]. Of note, the study from Portugal showed that type-2 and variant “f” were significantly…..”

9. Consider replacing the term “co-segregation” by the genetics term “linkage disequilibrium”?

Answer: We would like to thank the reviewer’s comments regarding this term. Nevertheless, in this case when we are specifically discussing variants in EBV genes, the correct term is cosegregation rather than “linkage disequilibrium”, meaning that the EBV with both variants will be present as separate entities in a determined population representing two different phenotypes. This is also in line with the terminology used by the literature regarding this topic (ref 23). Linkage disequilibrium refers a non-random allele association when referring to population genetics.

10. Line 199, do you mean “expression” as in RNA transcripts or do you mean genotype as in DNA variants?

Answer: We would like to thank the reviewer’s comments. By expression levels, the sentence highlights that both strains are different, meaning that they express different amounts of virulence factors such as cagA,  vacA, babB. This in turn implies higher transcripts levels of the mRNA encoding for these genes. This differential expression pattern between both strains is related to the pathogenicity of the bacteria regarding GC.

Reviewer 2 Report

In this manuscript, the authors review relationship between subtypes of EBV and gastric cancer. The authors mostly focused on two variations in the BamHI I fragment and XhoI site in the LMP1 gene, which have been reported for a long time. I find it worthwhile to report, but it needs some modifications and clarifications before publication.

Table 1: Is p value “0” correct?

Line 55: “whereas type-2 is the most common strain in Africa” must be wrong. Even in Africa, ratio of EBV-2 either in healthy people or BL patients is less than 50%.

Line 74: “viral chromosome” must read “viral genome”. Must be corrected throughout the text (eg, Legend for Fig1).

Line 74: what does “21,277 nucleotides” mean?

Line 152: What is “23/100.000 inhabitants”? Is it 23/100,000? If so, it does not sound very high as mentioned in the text.

Lines 193, 194: 100.000 means 100,000?

Authors need to explain in more details about “disrupted co-evolution” theory.

According to line 156-158, European ancestry is associated with lower GC ratio. However, Fig2 shows European EBV (BamHI-i/XhoI+) prevailed in GC in Chile. Why?

Author Response

Reviewer 2

Table 1: Is p-value “0” correct?

Answer: thanks for this comment. We have corrected p-value in Table 1

Line 55: “whereas type-2 is the most common strain in Africa” must be wrong. Even in Africa, a ratio of EBV-2 either in healthy people or BL patients is less than 50%.

Answer: we agree with this comment. We have changed to: “…..whereas type-2 is frequently found in Africa….”

Line 74: “viral chromosome” must read “viral genome”. Must be corrected throughout the text (eg, Legend for Fig1).

Answer: we have changed to “viral genome” as suggested

Line 74: what does “21,277 nucleotides” mean?

Answer: we delete this paragraph

Line 152: What is “23/100.000 inhabitants”? Is it 23/100,000? If so, it does not sound very high as mentioned in the text.

Answer: thanks for this observation. We accept your comment which has been modified through the text

Lines 193, 194: 100.000 means 100,000?

Answer: yes, this is correct. It has been changed in the text

Authors need to explain in more details about “disrupted co-evolution” theory.

Answer: Coevolution can promote a reduction in the antagonism between pathogen and host, as it shapes genome by genome interactions of both species. It has been proposed that the disruption of historical co-evolutionary relationships can explain many differences in disease outcomes. The latter is well described for H. pylori, as it normally has a largely innocuous and potentially symbiotic relationship with its host, which could be explained from a coevolutionary theory as it has a 50,000-year association with Homo sapiens. However, a fraction of individuals develops peptic ulcers and/or gastric cancer which can be explained by the disruption of the long-standing co-evolutionary relationship between them. This phenomenon is best exemplified by the study carried out by Kodaman [ref 84] in which participants were recruited from two Colombian populations with highly different rates of gastric cancer, despite a nearly universal prevalence of H. pylori infection in both. Main findings were that low-risk population was of admixed African, European, and Amerindian ancestry, whereas the high-risk population was mainly of Amerindian ancestry. The severity of gastric disease correlated with the proportion of African H. pylori ancestry in patients with primarily Amerindian ancestry, while patients with a large proportion of African human ancestry infected by African H. pylori strains had the best prognoses, consistent with ancestral coadaptation. The interaction between Amerindian human ancestry and African H. pylori ancestry accounted for the difference in disease risk between populations, whereas even the well-known virulence factor, CagA, did not. These findings are consistent with the idea that neither human nor H. pylori genetic variation confers susceptibility or virulence per se, but only in the context of mismatched co-evolution paradigm [ref 84]

According to line 156-158, European ancestry is associated with lower GC ratio. However, Fig2 shows European EBV (BamHI-i/XhoI+) prevailed in GC in Chile. Why?

Answer: Our statement is that EBV-associated GC is high in Amerindian populations infected with the European EBV (BamHI-i/XhoI+). Population in Chile is half Amerindian and half European, thus we speculated that EBV-associated GC in Chile should be mostly associated with the Mapuche ethnicity, a subtype of Amerindian ancestry. As stated in line 153 Mapuche ancestry is a well-established risk factor for GC in Chile.

Reviewer 3 Report

This short review provides valuable and unique information for studying prevalence of epidemic EBV-associated GC. The authors focuses on phylogenetic classification of EBV strains that would accounts for the mechanisms underlying EBV-GC in different populations. 

Author Response

we thank the reviewer for his comments and his approval.